# Executive function assessment: Adaptation of the Amsterdam executive function inventory using Spanish first-year university students from two knowledge areas

Elena Escolano-Pérez[1], Rita Pilar Romero-Galisteo[2]*, Jairo Rodríguez-Medina[3], Pablo Gálvez-Ruiz[4]

1 Department of Psychology and Sociology, University of Zaragoza, Zaragoza, Spain, 2 Department of Physiotherapy, University of Málaga, Málaga, Spain, 3 Department of Pedagogy, University of Valladolid, Valladolid, Spain, 4 Faculty of Law and Social Sciences, Valencian International University, Valencia, Spain

* rpromero@uma.es

## Abstract

### Objective

Many university students have difficulties in adapting to autonomous learning due to executive functioning deficits. In the Spanish university context, there is a lack of reliable validated instruments for the evaluation of executive functioning. In this sense, the aim of this research is to present the process of adaptation and validation of the Amsterdam Executive Function Inventory (AEFI) for the evaluation of executive functioning in the Spanish context.

### Methods

This study integrates two sequential processes: questionnaire translation and back-translation, and evaluation of the psychometric properties (exploratory and confirmatory factor analysis were conducted), reliability, validity and multigroup analysis to identify factorial invariance. An online questionnaire was used for data collection and R package *lavaan* software was administered to a sample of 519 first-year university students (270 females and 249 males).

### Results

The exploratory factor analysis evidenced an interna structure of three factors with adequate internal consistency (Cronbach's alpha higher than 0.70), endorsed in the confirmatory factor analysis that indicated an adequate goodness-of-fit-indexes for the model. The composite reliability showed values between 0.74 and 0.82, and the convergent (average variance extracted values ranged from 0.50 to 0.61) and discriminant validity were adequate. A multi-group-analysis showed the invariance factorial through the difference in the RMSEA, CFI and TLI index, performed both in the model comparison according to gender and academic disciplines.

**Data Availability Statement:** All relevant data are within the paper and its Supporting Information files.

**Funding:** This research was funded by the University of Zaragoza, Spain (Grant number: PIIDUZ_17_006). The funders had no role in study design, data collection and analysis, decision to publish, or preparation of the manuscript.

## Conclusion

The AEFI adapted for Spanish has practical implications for the management of university students, as it can facilitate the improvement of university policies designed to foster the development of executive functions, specifically in first-year students.

## Introduction

Going to university is a transition, and, as such, implies great academic, social, emotional, and personal change for which students are not always prepared [1]. For many students, this transition causes stress, anxiety, and insecurity, often leading to poor academic performance. It may also lead to students dropping out [2, 3]: the highest dropout rate, in fact, occurs in the first year of university, attributed to student difficulties adapting to a new teaching/learning environment [4–8].

Since learning in higher education is more autonomous and self-guided than in previous educational stages, first-year students are required to begin to coordinate and manage their own learning [9]. As autonomous and self-guided learners, students must therefore learn to set and develop strategies to achieve goals, plan, organize and prioritize time, materials and information, monitor their learning, and evaluate and adjust their activities to meet their self-set learning goals [6, 9–11]. Therefore, rather than focus exclusively on acquiring knowledge, students have to develop cognitive skills and processes that allow them to focus and monitor thoughts and actions to achieve their goals. Growing numbers of authors are consequently trying to explain self-regulated learning, and especially its difficulties, from the broader perspective of executive functioning [12–15].

Executive functioning, as a multidimensional construct defining a specific set of attention-regulation skills involved in conscious goal-directed problem solving, are particularly useful when it comes to resolving novel or complex problems often involving uncertainty, i.e., when no well-learned behaviors can be drawn on. Executive functioning involves independent and purposive behavior based on developing skills such as setting goals, planning strategies, organizing resources, focusing on selected aspects of particular problems, maintaining concentration, executing effective strategies, monitoring and evaluating progress, and making readjustments as needed [16, 17]. Self-regulated learning can thus be considered to be the contextualized application of the executive functions to the learning process [12–15, 18, 19]. Proper executive functioning is key to effective execution of academic tasks, productive learning, and academic success. Many studies report that executive functions are associated with successful learning and achievement [20–22], suggesting that students with poor executive functioning skills are likely to perform poorly in academic contexts [23, 24].

However, the importance of executive functions extends beyond the academic context. They are also essential for work, personal, family, and social success, since they enable thoughts and actions to be organized in a goal-directed way [16, 17]. Good executive functioning is also associated with improved health and quality of life [25]. Consequently, the executive functions play a vital role in everyday adaptation to everchanging environments, and so are key to achieving life-long wellbeing and success. Since the ultimate goal of contemporary education is to equip students with the necessary skills to become competent citizens in a changing world [26–28], the development of executive functions in students should be an educational priority and be part of university instruction.

Executive functions are malleable and consequently can be learned [29]. An appropriate period for executive skill training is adolescence, considered to begin at the onset of puberty

(around the age of 10 years) and to span the second decade of life up to the age of 17–20 years [30] or even 24 years [31]. In this developmental stage, important structural and functional changes occur in the dorsolateral prefrontal cortex, considered to underlie the development and refinement of the different skills associated with the executive functions. Brain structure specialization and functioning in adolescents ensures more complex, effective, and efficient executive operation [32–34].

Since university for most students starts at around the age of 18 years [35], the first year is key to enhancing the development of executive functions, as has been confirmed by various interventions [36]. Evaluating executive functions is necessary in order to be able to properly design individualized and effective interventions that meet the particular needs of each student. However, knowledge as to measurement as opposed to conceptual aspects of executive functions is limited [37], so, despite their relevance to academic success, very few instruments exist for the evaluation of executive functions in educational contexts. In addition, existing instruments usually consist of questionnaires completed by third parties (often teachers), so their reliability is questionable, especially because of bias originating in perceived social pressures, memory failures, and lack of familiarity and sensitivity regarding certain behaviors of subjects [38, 39]. Available instruments are also financially costly and, since they are generally composed of numerous items, they are time-consuming to implement. These drawbacks are further compounded by the fact that the instruments most frequently used to evaluate executive functions in the Spanish academic context are intended for compulsory educational levels [40, 41], not for university level. We consider it a matter of urgency that a brief free-of-charge instrument be available to evaluate executive functions in the Spanish university population.

The Amsterdam Executive Function Inventory (AEFI), initially developed by Van der Elst et al. [42] for adolescents aged15-18 years and later adapted by Baars et al. [43] for Dutch university students, is an easy-to-use tool for self-evaluation of executive functions. Since, as far as we are aware, no such evaluation instrument for executive functions exists for the Spanish university context with proper guarantees of reliability and validity, we implemented a transcultural adaptation of the AEFI instrument.

We translated, adapted, and validated the AEFI for first-year Spanish university students with a view to identifying possible executive functioning difficulties and designing interventions that respond to particular needs. Individualized attention is necessary in order to maximize the potential of each student (one of the objectives of the European Higher Education Area), since it is argued that the greatest wealth lies in intellectual capital [27]. Universities therefore have an important role to play in optimizing the potential of individuals and, consequently, increasing the intellectual capital and economic wealth of countries, given that education is key to a prosperous society [6]. This is particularly important in Spain, which has a higher university dropout rate than other European countries [27]. According to the latest data of the Spanish Ministry of Universities [44], the overall dropout rate in Spain is 33.2%, with 21.8% occurring in the first year of university. Furthermore, the percentage of Spanish students who complete their degree within the allotted time is low: just 36.2% of students graduate after four years (the standard duration for a Spanish undergraduate degree), with most students requiring 4.9 years to complete. Given the personal, family, economic and social consequences [5, 7], preventive measures are urgently required.

This study can make an important contribution in providing a validated instrument to identify executive function difficulties faced by first-year university students so that suitable interventions can be designed that will help them adjust to the learning requirements of the university context, reduce dropout rates and ultimately foster academic success [4].

## Methods

### Participants

An exploratory non-experimental design [45] was used to study a sample composed of 519 first-year university students, 52% (270) females and 48% (249) males, from the University of Zaragoza in Spain (a regional multi-campus–Huesca, Teruel and Zaragoza–face-to-face public university with diverse student cohorts). All the participants were informed of the research and signed the informed consent form.

Students from two different branches of knowledge were represented: 295 social sciences and law (53.6%), and 224 engineering and architecture (46.4%). Participants were mainly aged 18 (48.6%), 19 (15.2%) and 20 (13.3%) years old (M = 19.82; SD = 3.22; range 18–50); females were slightly older (M = 19.89; SD = 3.79) than males (M = 19.74; SD = 3.74).

### Instrument

To measure executive functions in our first-year university sample, we used the AEFI. Rather than the original 13-item scale developed by Van der Elst et al. [42], for this study we used the 10-item version by Baars et al. [43], as it is adapted to a university population of a similar age to our sample. The 10 items are grouped into three dimensions (reflecting three theoretical dimensions): attention (items 3, 6, 9), self-control (items 1, 5, 8, 10), and planning (items 2, 4, 7). Responses are scored on a 3-point Likert scale: 1 = not true, 2 = partly true, and 3 = true. We also included a block in the questionnaire to collect sociodemographic data (age, gender and degree course).

### Procedures

**Phase 1. AEFI translation and adaptation.** To adapt the AEFI questionnaire to Spanish, the translation/backtranslation procedure recommended in the specialized scientific literature was followed [46–48]. Two translators first independently translated the original version of the AEFI into Spanish. A group of 3 psychology experts (university professors with an average experience of 24.6 years in education psychology, learning and executive functions) then revised the items to produce a second version. The content validity of this second version was next analyzed by a group of 7 experts (university professors with an average experience of 20.5 years in higher education). Each of the 7 experts reviewed the items for 4 criteria: clarity (the item is easily understood, i.e., syntactically and semantically appropriate); coherence (the item has a logical relationship with the dimension or indicator it is measuring); relevance (the item is essential or important, i.e., it must be included), and sufficiency (items belonging to the same dimension are sufficient to measure that dimension). All items in each dimension were evaluated jointly. The scale proposed by Escobar and Cuervo [49] was used to evaluate the items regarding compliance with each of the 4 criteria: 1 = does not comply; 2 = complies poorly; 3 = complies moderately; 4 = fully complies. In the template to be completed by the education experts, guidance was provided regarding how to evaluate the criteria for each item; for example, in relation to the clarity criterion, a score of 1 meant that the item was not at all clear; 2, that the item required major syntactic and semantic modifications; 3, that the item required minor syntactic and semantic modifications; and 4, that the item was clear and syntactically and semantically appropriate. The template also contained a box for additional commentary by each expert.

To analyze interobserver agreement between the education experts, we calculated the Bang-diwala weighted concordance coefficients ($B_W^N$) [50] so as to graphically represent and obtain a measure of the strength of that agreement. To interpret the concordance coefficients, which

range from $B_W{}^N = 0$ (no agreement) to $B_W{}^N = 1$ (perfect agreement), Muñoz and Bangdiwala [51] propose the following agreement criteria: between 0.000 and 0.200, poor; between 0.201 and 0.400, weak; between 0.401 and 0.600, moderate; between 0.601 and 0.800, good; and above 0.801, excellent.

In Fig 1, the black, grey and white squares show observed agreement, partial agreement and no agreement, respectively. Bangdiwala concordance coefficient results for the 4 criteria were $B_W{}^N = 0.79$ (clarity), $B_W{}^N = 0.81$ (coherence), $B_W{}^N = 0.77$ (relevance), and $B_W{}^N = 0.73$

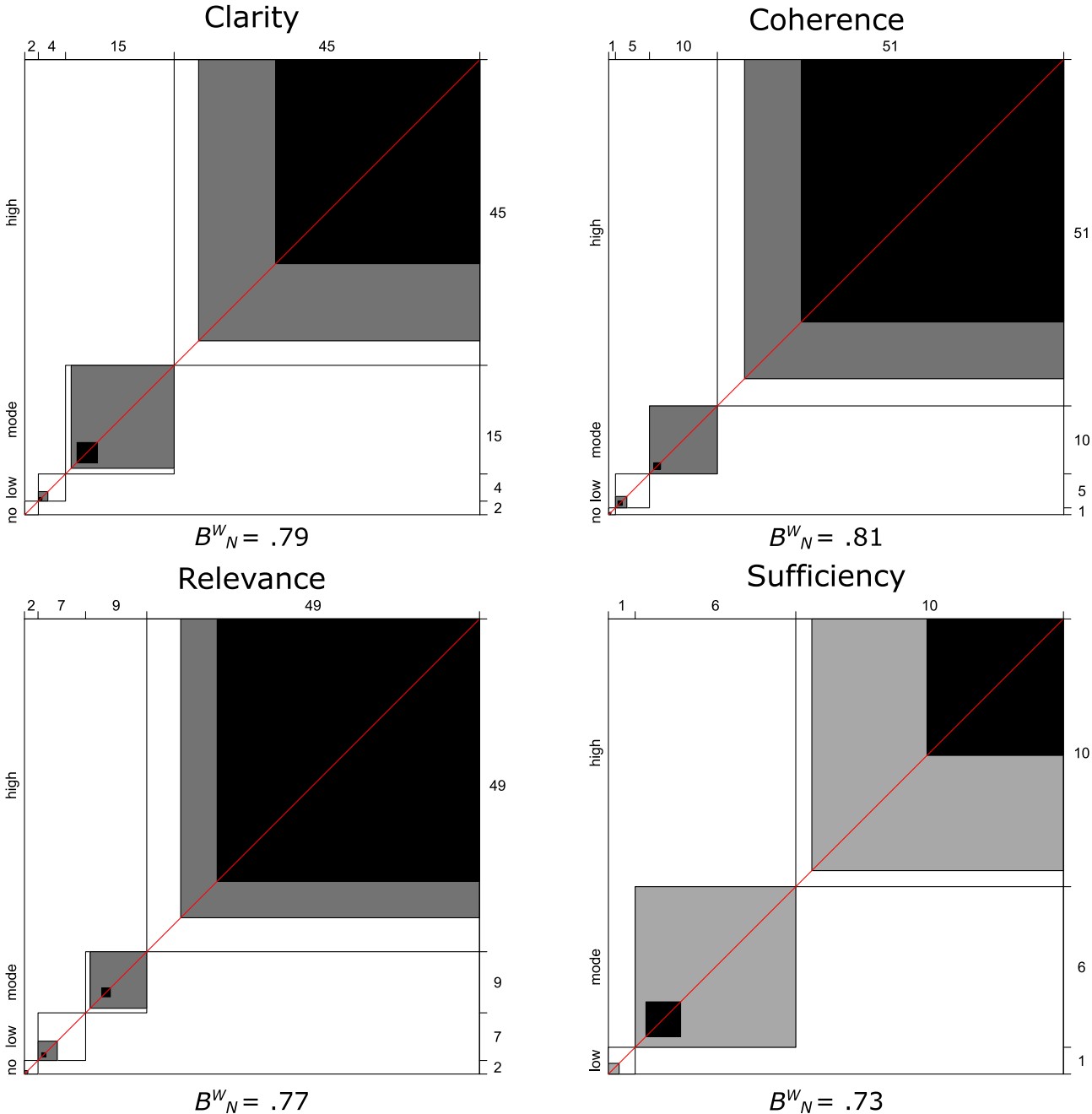

**Fig 1. Results of Bangdiwala interobserver agreement analysis regarding content validity.**

(sufficiency). Therefore, agreement was excellent for coherence and was good for clarity, relevance, and sufficiency. These rates of agreement suggest that the items overall can be considered appropriate for measuring executive functions in first-year Spanish university students.

Finally, two bilingual university professors (with an average experience of 15 years in higher education) independently backtranslated the validated version, then compared their versions to obtain a consensual translation. A bilingual teacher, expert in educational psychology (with 13 years of experience) next compared the consensual backtranslation to the original AEFI [43]. No further changes were proposed, and, hence, that was the version used in this research.

**Phase 2. Data collection.**   Data was collectioned was done during the months of in January and March 2018 after obtaining the approval by the University of Zaragoza Ethics Committee (code PI21/336). Students were sent an email with a link to the survey, which included an explanation of the purpose of the study and the voluntary nature of participation. Respondents were also guaranteed the anonymity and confidentiality of their data. Informed consent was obtained from all participants, who were asked for maximum sincerity and honesty and were informed that there were no right or wrong answers. Students were not offered any financial incentive for their participation in the study. The questionnaire took about 7–8 minutes to complete.

## Data analysis

Normal distribution (skewness and kurtosis) was assessed, and mean and standard deviation values were calculated. To ensure that that the questionnaire was valid, we conducted a variety of analyses. For two-factor analysis, the total sample was randomly divided into two.

For subsample $N_1$ (n = 253), the polychoric correlation matrix of the items [52] was analyzed with exploratory factor analysis (EFA), using principal components analysis (PCA) and Oblimin oblique rotation to identify the factorial structure of the model composed of the three dimensions (attention, self-control, and planning). To determine the number of factors, an optimized parallel analysis was implemented [53]. Exploratory graph analysis (EGA) [54] was also used to calculate the optimal number of factors that would summarize the dataset, after running the Kaiser-Meyer-Olkin (KMO) test of sampling adequacy and Bartlett test of sphericity. The internal reliability of the different dimensions was evaluated by Cronbach's alpha (α coefficient).

For subsample $N_2$ (n = 266), the polychoric correlation matrix of the items was analyzed using confirmatory factor analysis (CFA) and the robust weighted least-squares mean and variance estimator (WLSMV). No missing values were found in the data. Goodness-of-fit was tested using the chi-square/degrees-of-freedom ($\chi^2$/df) ratio. Because $\chi^2$ increases with sample size and df [55], multiple fit indexes were examined, namely, the comparative fit index (CFI), the Tucker-Lewis index (TLI), and the root mean square error of approximation (RMSEA). Considered acceptable for the purpose of this study were $\chi^2$/df ratios of less than 3 [56], and CFI and TLI values of more than 0.90 [52]; for RMSEA, values of less than 0.06 and 0.08 were rated as optimal and acceptable, respectively [57].

Factor loadings of the items were also scrutinized. Kline [58] recommends a relatively high standardized factor loading (e.g., 0.5) for an indicator to be retained. For the constructs, composite reliability (CR) was used to evaluate internal reliability, with values higher than 0.70 [52] rated as adequate, while average variance extracted (AVE) was used to test internal validity, with convergent validity demonstrated for values higher than 0.50 [52, 59].

Finally, to identify factorial invariance (FI), cross-validation procedures were used for the model with a multi-group strategy [60]. Invariance between the groups (by gender and knowledge areas) was tested in 4 progressively restrictive models [61]: configural invariance (all

parameters freely variable but with the structural model held constant), metric invariance (with constrained measurement weights), scalar invariance (with constrained factor loadings and thresholds), and strict invariance (all parameters equivalent across groups). Measurement invariance was assessed using three criteria proposed by Cheung and Rensvold [62] and Chen [63]: RMSEA change (ΔRMSEA) less than 0.01, CFI change (ΔCFI) greater than 0.01, and TLI change (ΔTLI) greater than 0.01. The maximum likelihood estimation method was used for all the models.

SPSS 21.0 was used to analyze the descriptive and some of the psychometric properties of the items, specifically, the exploratory factor analysis (EFA). The R package *lavaan* [64] was used to evaluate the measurement model through CFA and to verify the structure in each context (male and female groups).

## Results

### Preliminary analysis

The values for univariate skewness and kurtosis for all the variables were within conventional criteria for normality (-3 to 3 for skewness and -7 to 7 for kurtosis), as proposed by Finney and DiStefano [65]. The self-control dimension showed the lowest mean values, with three items below the theoretical response mean (1.5), while the two items with highest mean value were located in the planning dimension (items 2 and 4 with values of 2.26 and 2.36, respectively (Table 1).

### Exploratory factor analysis

The relevance of the EFA was demonstrated by a KMO value of .80, while Bartlett's test of sphericity was significant ($\chi^2(45) = 376.90$; $p < 0.001$). The optimized parallel analysis with 1000 random repetitions proposed an optimal solution with three factors. Using PCA and

**Table 1. Descriptive analysis of items.**

| Dimension and item | M | SD | US | K |
|---|---|---|---|---|
| Autocontrol / *Self-control* | | | | |
| 1. A menudo reacciono demasiado rápido. Hago o digo algo antes de que sea mi turno / *I often react too fast. I've done or said something before it is my turn* | 1.48 | 0.59 | 0.73 | -0.44 |
| 5. Comparado con otros, hablo mucho / *Compared to others, I talk a lot* | 1.75 | 0.74 | 0.45 | -1.07 |
| 8. Antes de actuar no pienso en las consecuencias de mis actos / *I do not consider the consequences before I act* | 1.42 | 0.62 | 1.21 | 0.37 |
| 10. Soy un bocazas, hablo más de la cuenta y de forma indiscreta / *I am a blabbermouth* | 1.23 | 0.49 | 2.04 | 3.40 |
| Planificación / *Planning* | | | | |
| 2. Me organizo bien, planifico adecuadamente lo que realizaré a lo largo del día / *I am well organized. For example, I am good at planning what I need to do during a day* | 2.26 | 0.67 | -0.35 | -0.79 |
| 4. Trabajo de manera muy ordenada / *My work is very tidy* | 2.36 | 0.64 | -0.48 | -0.67 |
| 7. Soy caótico y desorganizado / *I am chaotic or disorganized* | 1.44 | 0.63 | 1.12 | 0.14 |
| Atención / *Attention* | | | | |
| 3. No soy capaz de concentrarme en el mismo tema durante un largo periodo de tiempo / *I am not able to focus on the same topic for a long period of time* | 1.94 | 0.67 | 0.07 | -0.80 |
| 6. Me distraigo fácilmente / *I am easily distracted* | 2.03 | 0.68 | -0.04 | -0.86 |
| 9. Mi pensamiento se dispersa fácilmente / *My thoughts easily wander* | 1.92 | 0.67 | 0.10 | -0.81 |

*Note*. Italics, original items of Baars et al. [43]

M = mean; SD = standard deviation; US = univariate skewness; K = kurtosis

**Table 2. Factor loadings, Z-values, composite reliability and average variance extracted.**

| Dimension and item | EFA loadings | CFA loadings | Z-values | CR | AVE |
|---|---|---|---|---|---|
| Autocontrol / *Self-control* | | | | 0.77 | 0.53 |
| 1. A menudo reacciono demasiado rápido. Hago o digo algo antes de que sea mi turno / *I often react too fast. I've done or said something before it is my turn* | 0.657 | 0.695 | 10.02 | | |
| 5. Comparado con otros, hablo mucho / *Compared to others, I talk a lot* | 0.683 | 0.713 | 10.37 | | |
| 10. Soy un bocazas, hablo más de la cuenta y de forma indiscreta / *I am a blabbermouth* | 0.619 | 0.774 | 6.39 | | |
| Planificación / *Planning* | | | | 0.74 | 0.50 |
| 2. Me organizo bien, planifico adecuadamente lo que realizaré a lo largo del día / *I am well organized. For example, I am good at planning what I need to do during a day* | 0.754 | 0.693 | 10.55 | | |
| 4. Trabajo de manera muy ordenada / *My work is very tidy* | 0.771 | 0.730 | 8.56 | | |
| 7. Soy caótico y desorganizado / *I am chaotic or disorganized* | 0.777 | 0.676 | 7.09 | | |
| Atención / *Attention* | | | | 0.82 | 0.61 |
| 3. No soy capaz de concentrarme en el mismo tema durante un largo periodo de tiempo / *I am not able to focus on the same topic for a long period of time* | 0.660 | 0.698 | 12.05 | | |
| 6. Me distraigo fácilmente / *I am easily distracted* | 0.818 | 0.761 | 7.79 | | |
| 9. Mi pensamiento se dispersa fácilmente / *My thoughts easily wander* | 0.794 | 0.873 | 6.03 | | |

*Note*. Italics, original items of Baars et al. (2015); EFA = exploratory factor analysis; CFA = confirmatory factor analysis; CR = composite reliability; AVE = average variance extracted.

oblique Oblimin rotation, a structure was obtained that explained 60% of variance; all items had a factor loading above 0.60 except item 8 (self-control dimension), whose loading was 0.47 (Table 2). The EGA suggested a solution of three clusters (Fig 2), corresponding fully with the three instrument dimensions of attention, self-control, and planning. In this figure nodes represent items, green lines represent positive connections and red lines represent negative connections between items; the three factors or clusters are colored.

Internal reliability and internal validity were adequate ($\alpha$ = 0.70, 0.53, and 0.71 for the attention, self-control and planning dimensions, respectively), and all corrected item-scale correlations were above 0.30, except for item 8 (self-control dimension), whose correlation was 0.26.

## Confirmatory factor analysis

The goodness-of-fit indexes from the CFA indicated that the model fitted the data well. The $\chi^2$/df ratio of 1.73 was below 3 ($\chi^2(32)$ = 55.58; $p < 0.01$). Values for the other fit indexes were CFI = 0.96 and TLI = 0.95, while the RMSEA index was 0.049, with values of 0.034 and 0.064 for the minimum and maximum confidence interval (CI) of 90%, respectively.

Factor loadings (Table 2) between the latent and observable variables demonstrated standardized values above 0.50, and also adequate individual reliability ($R^2 \geq 0.25$) [66], except for item 8 (self-control dimension), which was eliminated due to its value being lower than the established cutoff point ($R^2 = 12$). After these modifications, model fit was found to be satisfactory ($\chi^2(24)$ = 41.36; $p = 0.015$; $\chi^2$/df = 1.72; CFI = 0.97; TLI = 0.95; RMSEA = 0.037; 90% CI [0.016, 0.056]).

The CR values were above the recommended 0.70 [52] for each of the constructs, and the AVE was greater than 0.50 [52], indicating adequate convergent validity [59] (Table 2).

## Factorial invariance analysis

The focus with FI was on invariance in the measurement instrument. The first step considered the model for individual subsamples, showing a good fit for both the male group ($\chi^2(24)$ =

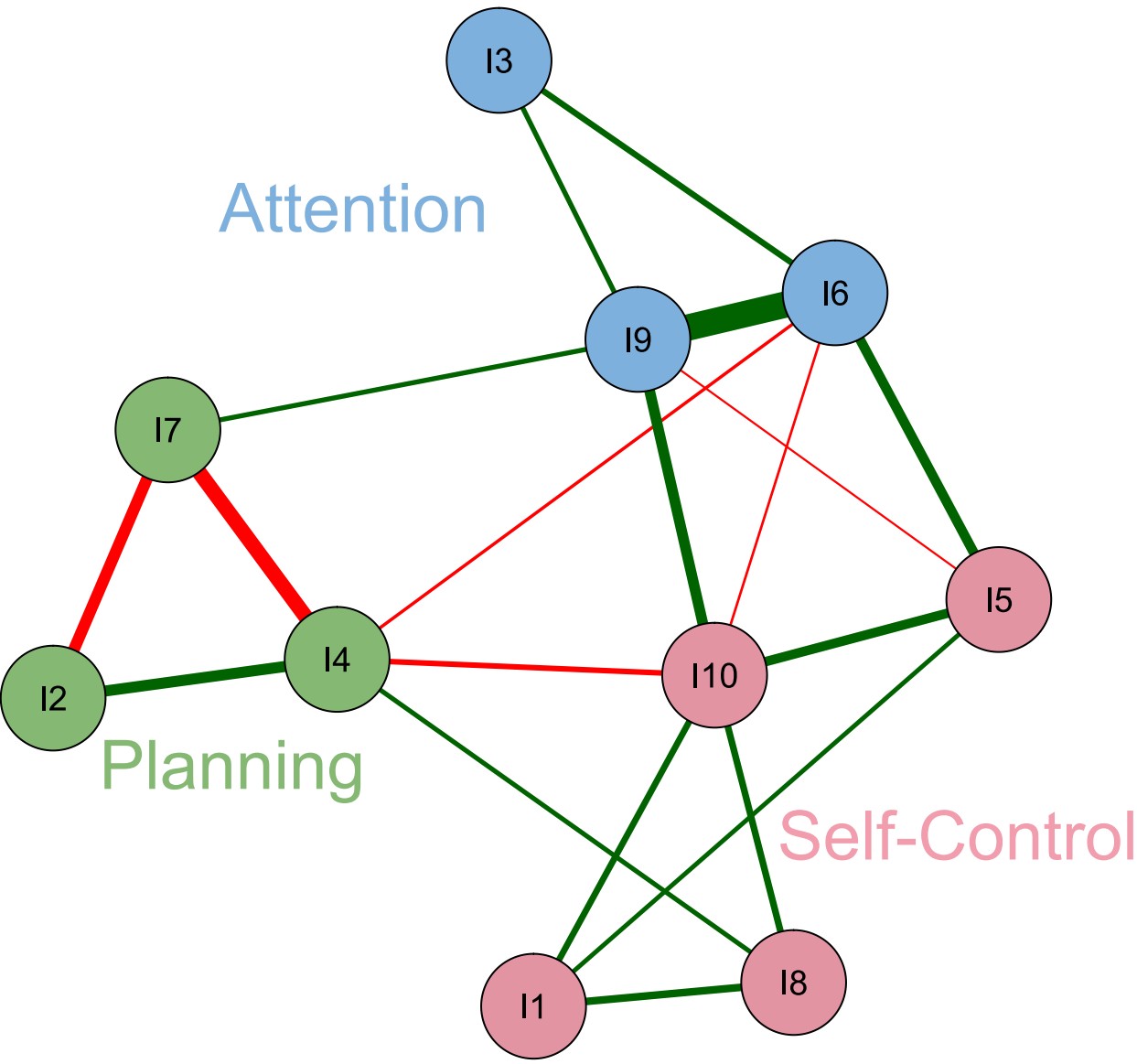

**Fig 2. Network of partial correlations estimated during exploratory graph analysis showing three latent dimensions.**

41.77; $p$ = 0.014; $\chi^2$/df = 1.74; CFI = 0.96; TLI = 0.95; RMSEA = 0.055; 90% CI[0.025, 0.082]) and for the female group ($\chi^2$(24) = 33.87; $p$ = .087; $\chi^2$/df = 1.41; CFI = 0.98; TLI = 0.97; RMSEA = 0.039; 90% CI[0.016, 0.067]).

FI across gender was verified through an unconstrained model and a model with constrained measurement weights [61]. The configural invariance model, which corresponded to invariance in the basic measurement model configuration, proposed the same factor loading pattern for the male and female groups. The reference model, therefore, tested the hypothesis that the same general factor loading pattern remained constant for both groups. The indexes showed an adequate fit of the configural invariance model, which supports the assumption that the items represented the same underlying construct in both groups. Table 3 shows measurement invariance by gender according to the data analysis criteria. Adequate goodness-of-fit indexes were obtained for the unconstrained model (CFI > 0.90; TLI > 0.90; RMSEA<

**Table 3. Measurement invariance by gender.**

| Model | χ² (df) | Δχ²(Δdf) | p | RMSEA | ΔRMSEA | CFI | ΔCFI | TLI | ΔTLI |
|---|---|---|---|---|---|---|---|---|---|
| Configural | 64.65 (48) | - | - | 0.037 | - | 0.967 | - | 0.961 | - |
| Metric | 65.83 (54) | 1.18 (6) | 0.956 | 0.029 | 0.008 | 0.975 | 0.008 | 0.967 | 0.006 |
| Scalar | 69.59 (60) | 3.76 (6) | 0.522 | 0.025 | 0.004 | 0.974 | 0.001 | 0.968 | 0.001 |
| Strict | 80.08 (69) | 10.49 (9) | 0.235 | 0.025 | 0 | 0.971 | 0.003 | 0.970 | 0.002 |

*Note*. RMSEA = root mean square error of approximation; CFI = comparative fit index; TLI = Tucker-Lewis index; df = degrees of freedom; Δ(CFI, TLI, RMSEA) = change in fit with respect to the previous least restrictive model. Configural invariance (for identification purposes): one marker variable per factor set to 1, unique variance of marker variables set to 1; unique variance of first group set to 1, factor mean of first group set to 0. Metric invariance: loadings constrained to be equal across groups. Scalar invariance: factor loadings and thresholds constrained to be equal across groups. Strict invariance: all unique variances of all groups set to 1.

0.05), indicating that the participants in the two subsamples used the same conceptual framework to respond to the items in the scale, thereby confirming configural invariance [62, 67]. Consequently, it can be concluded that factors for the items in terms of both number and loading patterns were similar for males and for females. To check for another level of invariance, the configural invariance model was compared with the metric invariance model, which added the restriction that the factor loadings were identical in both groups. Again, factors for the items in terms of both number and loading patterns were similar for males and for females, with differences in RMSEA, CFI and TLI lower than 0.01 (ΔRMSEA = 0.008; ΔCFI = 0.008; ΔTLI = 0.006).

The evidence of metric invariance indicated that the contribution of each item to the latent variables remained constant in the different groups and, therefore, that it was appropriate to compare the groups in terms of latent variable variances and covariances using a scalar invariance test, which allows comparisons between group means. The scalar invariance model added the restriction that, for each indicator, the thresholds were invariant in both groups. As can be seen in Table 3, the scalar invariance model did not significantly worsen the fit of the less restrictive metric invariance model (ΔRMSEA = 0.004; ΔCFI = 0.001; ΔTLI = 0.001). Finally, the strict invariance model tested the null hypothesis that error variances for each item were equivalent for males and for females. Since the between-group differences in the observed variables were attributable to differences in the latent common factors, the between-group differences in the manifest variables had to be the result of differences in the latent factors. Again, as can be seen in Table 3, this model did not significantly worsen the fit compared to the less restrictive scalar invariance model (ΔRMSEA = 0; ΔCFI = 0.003; ΔTLI = 0.002).

The same procedure as used for gender was used to check invariance between participants from different academic disciplines (social sciences and law versus engineering and architecture), with Table 4 reporting evidence of strict invariance.

Overall, our results show the usefulness of the validated AEFI scale to evaluate executive functioning in groups of students of different gender and from different academic disciplines.

## Discussion

For the purpose of evaluating executive functions in first-year university students in Spain, we translated and adapted the AEFI and validated it by analyzing its psychometric properties and factorial structure. We used the 10-item AEFI adapted for first-year Dutch university students by Baars et al. [43], based on the original 13-item scale by Van der Elst et al. [42]. The items and dimensions in the original scale have been demonstrated to be internally reliable, facilitating adaptation to different contexts. Use of the original scale has also been extended to

**Table 4. Measurement invariance across disciplines.**

| Model | χ²(df) | Δχ²(Δdf) | p | RMSEA | ΔRMSEA | CFI | ΔCFI | TLI | ΔTLI |
|---|---|---|---|---|---|---|---|---|---|
| Configural | 62.27 (48) | - | - | 0.039 | - | 0.968 | - | 0.952 | - |
| Metric | 74.93 (54) | 6.25 (6) | 0.395 | 0.039 | 0 | 0.969 | 0.001 | 0.958 | 0.006 |
| Scalar | 78.85 (60) | 5.54 (6) | 0.476 | 0.035 | 0.004 | 0.967 | 0.002 | 0.960 | 0.008 |
| Strict | 91.45 (69) | 13.46 (9) | 0.143 | 0.035 | 0 | 0.963 | 0.004 | 0.961 | 0.001 |

*Note*. RMSEA = root mean square error of approximation; CFI = comparative fit index; TLI = Tucker-Lewis index; df = degrees of freedom; Δ(CFI, TLI, RMSEA) = change in fit with respect to the previous least restrictive model. Configural invariance (for identification purposes): one marker variable per factor set to 1, unique variance of marker variables set to 1; unique variance of first group set to 1, factor mean of first group set to 0. Metric invariance: loadings constrained to be equal across groups. Scalar invariance: factor loadings and thresholds constrained to be equal across groups. Strict invariance: all unique variances of all groups set to 1.

children aged 9–12 years [68], although psychometric properties (reliability and validity) have not been thoroughly verified.

In our study based on the Baars et al. [43] version, we conducted an exhaustive analytical process that provides sufficient evidence of validity and reliability regarding interpretation of the instrument scores. Our results reflect adequate scale properties, supporting the factorial structure of the AEFI for Spain. However, the analyses indicate the need to exclude item 8, corresponding to the self-control dimension (*"Antes de actuar no pienso en las consecuencias de mis actos"/I do not consider the consequences before I act*), as it shows low loading in both the EFA and CFA analyses. Considered adequate were the goodness-of-fit of the different indexes, and also factorial invariance according to gender and knowledge domains, thereby confirming the stability of the factorial structure.

Our content validity analyses are an important contribution to both the original Van der Elst et al. [42] and the modified Baars et al. [43] versions of the AEFI, as no such analyses were performed by those authors. Thus, going beyond the translation of the scale to Spanish, our experts considered the items to comply with the four key criteria of clarity, coherence, relevance, and sufficiency. Another important contribution is that our adaptation has been demonstrated to be invariant, not only by gender, but also for two knowledge domains (social sciences and law versus engineering and architecture), whereas the 10-item version [43] was only demonstrated for students of applied science.

Several reasons underpin the value and usefulness of the validation of the AEFI as described in this study. Major advantages of the AEFI is that it is both brief and free of charge, unlike the lengthy (80-item) and costly Behavior Rating Inventory of Executive Function (BRIEF) [69], the most widely used scale for the evaluation of executive functions in adolescence [70]. BRIEF versions not available in Spain are the self-reported BRIEF (BRIEF-SR) for adolescents (only versions for parents, teachers, or other educational professionals familiar with the adolescent) or the adult version (BRIEF-A). It should be noted that the BRIEF (in all its versions) was designed for the clinical evaluation and treatment of executive control functioning problems, i.e., the target population belongs to a clinical rather than educational context. Another instrument to evaluate executive functions in adolescents is the recently developed Teenage Executive Functioning Inventory (TEXI) [71]. Like the AEFI, it is short (20 items) and free of charge. It also has both self-reported and parent/teacher report versions, although as yet no version for Spanish is available.

The brevity (9 items) and validity (good psychometric properties) of our AEFI for Spanish first-year university students are highly relevant to its practical application. The fact that it is free is also important, in view of the current economic difficulties faced by Spanish universities and research in the social sciences and education. In Spain, public spending (in terms of gross

domestic product) earmarked for science in general (1.24%) is well below the European average (2.12%). Therefore, the availability of free instruments that facilitate the implementation of strategies aimed at improving education and training is crucial.

It should be mentioned that the brevity of the AEFI is in line with executive functions as conceptualized by Anderson et al. [72], whose model guided its development. Both the AEFI versions [42, 43] conceptualize executive functions in three separable but integrated dimensions: attention, self-control, and planning. Other theoretical and empirical models of executive functions exist in the literature that reflect numerous components of executive functions. BRIEF, for instance, because it is constructed from a much more complex theoretical model, evaluates far more executive components and includes both cognitive and affective components; however, this makes it far more complex to administer.

This study contributes to meeting the lack of an instrument in Spanish to evaluate the executive functions in university students that are crucial to learning and academic success. It also has practical implications for the management of university students, as it can facilitate the improvement of university policies designed to foster the development of executive functions, specifically in first-year students. In this sense, university policies could be developed to improve the stress that often affects these students [2, 3, 73] and, on the other hand, address this issue in teacher training programmes [74].

## Limitations and future studies

The main limitation on this study was the impossibility of calculating convergent validity, given the lack of scales in Spanish for first-year university students and the lack of sources with which to compare our results. Another limitation was the relatively small sample from a single university, when ideally the sample should be larger and representative of other national university populations, not to mention a broader selection of knowledge domains. For this reason, we underline the importance of conducting other studies that would allow comparisons of executive functioning in students from all branches of knowledge and from different kinds of universities (public, private, offline, and online), and also to check the psychometric properties of the AEFI in university students other than first-years, i.e., with some experience of university learning, so as identify possible differences in executive functioning in different years. This kind of validation would also make it possible to carry out longitudinal studies to verify whether executive functioning improves as students advance in their university studies, and to assess the effectiveness of interventions with first-year students.

Finally, since the translated, adapted and validated AEFI has the same limitations as occur with all self-reported data (e.g., perceived social pressures), it may be interesting to supplement surveys with systematic observation.

## Conclusions

We conclude that our Spanish version of the AEFI has good psychometric properties and might become a useful tool for the evaluation of executive functions in first-year Spanish university students. The detection of students with executive functioning difficulties will facilitate the design of interventions that should contribute to greater academic and career success in students.

## Author Contributions

**Conceptualization:** Elena Escolano-Pérez, Rita Pilar Romero-Galisteo, Jairo Rodríguez-Medina.

**Data curation:** Pablo Gálvez-Ruiz.

**Formal analysis:** Jairo Rodríguez-Medina, Pablo Gálvez-Ruiz.

**Funding acquisition:** Elena Escolano-Pérez.

**Methodology:** Pablo Gálvez-Ruiz.

**Software:** Jairo Rodríguez-Medina, Pablo Gálvez-Ruiz.

**Validation:** Pablo Gálvez-Ruiz.

**Writing – review & editing:** Elena Escolano-Pérez, Rita Pilar Romero-Galisteo, Pablo Gálvez-Ruiz.

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
