## [Decision Letter · Decision Letter 0]

11 Jul 2022

PONE-D-22-11155EXECUTIVE FUNCTION ASSESSMENT: ADAPTATION OF THE AMSTERDAM EXECUTIVE FUNCTION INVENTORY USING SPANISH FIRST-YEAR UNIVERSITY STUDENTS FROM TWO KNOWLEDGE AREASPLOS ONE

Dear Dr. Romero-Galisteo,

Thank you for submitting your manuscript to PLOS ONE. After careful consideration, we feel that it has merit but does not fully meet PLOS ONE’s publication criteria as it currently stands. Therefore, we invite you to submit a revised version of the manuscript that addresses the points raised during the review process.

We look forward to receiving your revised manuscript.

Kind regards,

Gabriel G. De La Torre

Academic Editor

PLOS ONE

Journal Requirements:.

“This research was funded by the University of Zaragoza, Spain (Grant number: PIIDUZ_17_006)”

6. Please review your reference list to ensure that it is complete and correct. If you have cited papers that have been retracted, please include the rationale for doing so in the manuscript text, or remove these references and replace them with relevant current references. Any changes to the reference list should be mentioned in the rebuttal letter that accompanies your revised manuscript. If you need to cite a retracted article, indicate the article’s retracted status in the References list and also include a citation and full reference for the retraction notice

Reviewers' comments:

Reviewer's Responses to Questions

**Comments to the Author**

1. Is the manuscript technically sound, and do the data support the conclusions?

Reviewer #1: Yes

Reviewer #2: Yes

2. Has the statistical analysis been performed appropriately and rigorously? 

Reviewer #1: Yes

Reviewer #2: Yes

3. Have the authors made all data underlying the findings in their manuscript fully available?

Reviewer #1: Yes

Reviewer #2: Yes

4. Is the manuscript presented in an intelligible fashion and written in standard English?

Reviewer #1: Yes

Reviewer #2: Yes

5. Review Comments to the Author

Reviewer #1: I would like to congratulate authors for this research and for the manuscript, since a valuable instrument has been adapted and validated with methodological rigor.

Nevertheless, I would like to make some comments:

- Even when in general the manuscript is written in stantard English and in an intelligible fashion, I would liket to ask authors to review this aspect, since many errors have been detected (i.e. lines 218-219, or line 229).

- Data analysis: I still do not have clear which statistical program has been used for the exploratory factor analysis, and it would be interesting to clarify this point in order to facilitate replicability.

- Results: item 8 is not included in tables 1 and 2. In this respect, Table 2 is referenced in line 286, but then this table doesn't include the item.

I do not have any other important comment. Thank you.

Reviewer #2: Comments on PLOS ONE: Executive Function Assessment: Adaptation of The Amsterdam Executive Function Inventory Using Spanish First-Year University Students from Two Knowledge Areas

The topic is interesting and important to have the Spanish version of “Executive Function Assessment Inventory”. Besides, as you have stated well in the introduction section of your manuscript, for the Spanish researchers it could be good opportunities to use for the stated purposes and ways.

The way you have approached and gone through the process during your studying is good. The sampling, composition and data analyses procedures are interesting.

Minor comments

1.Instead of using women and men it is better to use females and males, because in different cultures there may be

different meanings (women or men go to adult people, particularly in some societies married individuals).

2.In the line 154-156 regarding age courts of research participants 18 years 48.6%, 19 years 15.2% and 20 years 13.3%

are indicated. But what about the rest (22.9%)? Is it insignificant or unknown? You need to clarify the reason why you

have overlooked it.

3.Line 219- “approval form university of XXX ethics committee” it says. What is the need to hide the name of the

university? I think it must be disclosed unless there is conflict of interest? By the way is there XXX university? In your

country if there is that is correct.

4.Table 1: Descriptive of items, you have presented the data only in the form of a table but it requires describing what the

analysis in the table means shortly for your readers immediately under the table.

5.Besides in the same table the English version of item 3 (in attention section) states that “I am no table to focus….” in

other places also there is a minor error in the same item you need to correct it (I am not able to focus…...)

6.Exploratory factor analysis (page 12) if you are able to show the data with some form of pictures or charts for easy

understanding by your readers. Otherwise, if you have figures or charts in the appendix section you need to give clues

for your readers where to get such types of additional/supplementary information. Unless and otherwise, it seems

something that is suddenly presented, or else it seems as a format/style of summary or discussion of the results.

7.The same page (page 12) Exploratory factor analysis as well as Confirmatory factor analysis seems as if they have

suddenly come into these places. First let you present your data for example in the form of a table, then after continuing

with descriptions, this will be natural sequencing in presentation. In short, it is good if your data analysis/interpretation c

comes after the presentations of data in the form of a table or/ figure!! Of course, since figures need to be presented

separately according to the POLOS ONE formatting style you may refer to the figure and where to find the figure for your

readers (you can say as indicated in figure ___ (see appendix section) etc.)

8.The discussion section is not done thoroughly, besides, it is advised to manage the conclusions section separately from

discussion if you need to have a conclusion section.

9. I need clarification about the figures indicated in the appendix. It required some sort of explanation. Generally as a principle, it is supposed that a figure explains itself more than many words, however here I wasn't able to understand the messages of the figures. Besides, it will be good to give names of the figures and some remarks so that readers are able to understand them more easily.

10. line 150-151 written that "An exploratory non-experimental design [45] was used to study a sample 151 composed of

519 first-year university students", what does 45 indicate? Is it an author? If that is so, what is the need? Has the author (45) given you the number of samples that you have to use?

11. The sampling in line 154-155 What are your rational/methods to determine your sampling (295 social sciences and law (53.6%), and 224 engineering and architecture (46.4%)? Method should be clear and logical.

12. line 148 Materials and method- what does materials mean for you? Have you used any material in conducting this study? My answer is no? Then better to say Methods only ( if you use laboratory/experimental research you need to use materials where you have used human beings as research participants, that is all).

13. Even line 159 "Measures" do you think appropriate here? I think instruments seem more sound because they are accustomed to most social and behavioural research.

6. PLOS authors have the option to publish the peer review history of their article (what does this mean?). If published, this will include your full peer review and any attached files.

Reviewer #1: No

Reviewer #2: **Yes: **Dereje Adefris

---

## [Author Response · Author response to Decision Letter 0]

25 Jul 2022

ITEMIZED LIST OF THE REVIEWER COMMENT

PLOS ONE

Submission D-22-11155

Title: Executive function assessment: adaptation of the Amsterdam Executive Function Inventory using Spanish first-year University students from two knowledge areas

Version: 1

Date: July 2022

Editor Comments

Thank you for submitting your manuscript to PLOS ONE. After careful consideration, we feel that it has merit but does not fully meet PLOS ONE’s publication criteria as it currently stands. Therefore, we invite you to submit a revised version of the manuscript that addresses the points raised during the review process.

• Authors: 

Dear Editor,

Please, find a revision of our manuscript entitled Executive function assessment: adaptation of the Amsterdam Executive Function Inventory using Spanish first-year University students from two knowledge areas.

We would like to thank the Reviewers for their thoughtful and constructive comments. We have considered all suggestions, and have incorporated them into the revised manuscript. We believe our manuscript is stronger as a result of the modifications. An itemized point-by-point response to the Reviewers’ comments is presented below.

REVIEWER 1

I would like to congratulate authors for this research and for the manuscript, since a valuable instrument has been adapted and validated with methodological rigor.

• Authors: We would like to thank you the anonymous reviewer for his/her positive comments of our manuscript.

Nevertheless, I would like to make some comments:

- Even when in general the manuscript is written in stantard English and in an intelligible fashion, I would liket to ask authors to review this aspect, since many errors have been detected (i.e. lines 218-219, or line 229).

• Authors: We thank the reviewer comments. According to the suggestions, the new version of the manuscript has been completely revised and we have the revision certificate. We apologize for any errors detected.

- Data analysis: I still do not have clear which statistical program has been used for the exploratory factor analysis, and it would be interesting to clarify this point in order to facilitate replicability.

• Authors: We are grateful to your helpful comments. The software used for the statistical analyzes is exposed in the final part of the "Data analysis" section. We have added a comment to specify the statistical software used to perform the exploratory factor analysis (new version: line 268).

- Results: item 8 is not included in tables 1 and 2. In this respect, Table 2 is referenced in line 286, but then this table doesn't include the item.

• Authors: We thank the reviewer for raising this important issue. Indeed, item 8 does not appear in tables 1 and 2. It is an error in the case of table 1, where the initial descriptive data of the questionnaire used are exposed, and therefore, item 8 must be included here (item 8 can be checked in the new version of table 1 and the statistical results obtained). However, it does not appear in Table 2 because this item does not meet various methodological criteria considered: factorial weight in the exploratory factorial analysis, internal consistency analysis, and factorial weights in the confirmatory factorial analysis. In lines 307-310 (new version) we indicate the elimination of item 8. 

I do not have any other important comment. Thank you.

• Authors: We sincerely appreciate the comments made by reviewer 1. These are undoubtedly good insights that help improve the final document.

REVIEWER 2

The topic is interesting and important to have the Spanish version of “Executive Function Assessment Inventory”. Besides, as you have stated well in the introduction section of your manuscript, for the Spanish researchers it could be good opportunities to use for the stated purposes and ways. The way you have approached and gone through the process during your studying is good. The sampling, composition and data analyses procedures are interesting.

• Authors: We really appreciate for your great work for our manuscript and your suggestions, which have helped us much to improve its quality.

Minor comments

1.Instead of using women and men it is better to use females and males, because in different cultures there may be different meanings (women or men go to adult people, particularly in some societies married individuals).

• Authors: Thanks a lot for the suggestion. We have reviewed the entire manuscript to address this suggestion.

2.In the line 154-156 regarding age courts of research participants 18 years 48.6%, 19 years 15.2% and 20 years 13.3% are indicated. But what about the rest (22.9%)? Is it insignificant or unknown? You need to clarify the reason why you have overlooked it.

• Authors: We understand the reviewer's reflection. Lines 155-158 (new version) specifically indicate the main age of the participants, which is mainly between 18 and 20 years old. The age range is reported below, specifically, between 18 and 50 years. It is important to consider that the study is aimed at first-year university students, the main reason why 77.1% are between 18 and 20 years old. The rest of the ages in the specified range have a large dispersion, which is why we had decided not to specify them.

3.Line 219- “approval form university of XXX ethics committee” it says. What is the need to hide the name of the university? I think it must be disclosed unless there is conflict of interest? By the way is there XXX university? In your country if there is that is correct.

• Authors: The information about the university has been eliminated in version 1 of the manuscript in order to avoid (or blinding) any information about the authors. The information requested by the reviewer is included in the new version (line 219).

4.Table 1: Descriptive of items, you have presented the data only in the form of a table but it requires describing what the analysis in the table means shortly for your readers immediately under the table.

• Authors: Thank you for this comment. Following the suggestion of the reviewer, a brief explanation of the results obtained from the mean value of the items has been included (new version: lines 275-278), since the information regarding normality was already exposed.

5.Besides in the same table the English version of item 3 (in attention section) states that “I am no table to focus….” in other places also there is a minor error in the same item you need to correct it (I am not able to focus…...).

• Authors: We sincerely apologize for this error, we have not perceived this detail in our revisions prior to submitting the manuscript. We have eliminated this error in Tables 1 and 2.

6.Exploratory factor analysis (page 12) if you are able to show the data with some form of pictures or charts for easy understanding by your readers. Otherwise, if you have figures or charts in the appendix section you need to give clues for your readers where to get such types of additional/supplementary information. Unless and otherwise, it seems something that is suddenly presented, or else it seems as a format/style of summary or discussion of the results.

• Authors: Thank you for your comments. For the exploratory factor analysis, the authors include in the corresponding section the main results obtained and rely on Figure 2, indicated on line 289. In this analysis we have added a brief explanation of the meaning of Figure 2, accepting the reviewer's suggestion 9. In addition, in the section "Data analysis" the different statistical analyses to be performed are presented, specifically in relation to the exploratory factor analysis, in lines 232-239.

7.The same page (page 12) Exploratory factor analysis as well as Confirmatory factor analysis seems as if they have suddenly come into these places. First let you present your data for example in the form of a table, then after continuing with descriptions, this will be natural sequencing in presentation. In short, it is good if your data analysis/interpretation comes after the presentations of data in the form of a table or/figure!! Of course, since figures need to be presented separately according to the PLOS ONE formatting style you may refer to the figure and where to find the figure for your readers (you can say as indicated in figure ____ (see appendix section) etc.).

• Authors: Thanks for your comments. We consider that the sequence is correctly written, that is, the results obtained in the exploratory factor analysis are presented and Figure 2 is included, which supports the three-factor solution. Next, the results obtained for the confirmatory factorial analysis are written, using the support of Table 2, which includes the factorial weights of both analyses, together with the results of composite reliability and average variance extracted. Table 2 contains all the information from the factorial analyzes and thus avoids including 2 very similar tables.

8.The discussion section is not done thoroughly, besides, it is advised to manage the conclusions section separately from discussion if you need to have a conclusion section.

• Authors: We are grateful for your suggestion. Most of the discussion section revolves around the main objective of our study, which is the cross-cultural adaptation of a measurement instrument. Even so, we have added a few brief sentences at the end (page 20, lines 472-478), where, in order not to make this section too long, we have also added 2 new bibliographical citations. We believe that the discussion section touches on important aspects in relation to the other studies published on this topic. We strongly agree with you that adding a section on conclusions would improve the quality of our work.

9. I need clarification about the figures indicated in the appendix. It required some sort of explanation. Generally as a principle, it is supposed that a figure explains itself more than many words, however here I wasn't able to understand the messages of the figures. Besides, it will be good to give names of the figures and some remarks so that readers are able to understand them more easily.

• Authors: We thank the reviewer for his/her appreciation. The figures are found in the appendix following the rules for authors of PLOS ONE. However, the content of figure 1 is exposed on lines 201-207, while in the case of figure 2, it is exposed on lines 290 -292. However, we have included a brief explanation about Figure 2 on lines 288-292 (new version).

10. line 150-151 written that "An exploratory non-experimental design [45] was used to study a sample 151 composed of 519 first-year university students", what does 45 indicate? Is it an author? If that is so, what is the need? Has the author (45) given you the number of samples that you have to use?

• Authors: We thank the reviewer for his/her interest in this issue. The number 45 indicates the reference number of Ato et al. (2013), entitled “A classification system for research designs in psychology”. This work elaborates a conceptual framework and develops some basic principles to promote a classification system of research designs based on three strategies (manipulative, associative and descriptive). Non-experimental studies are included within the descriptive strategy, which according to the authors are those that do not meet any of the two basic criteria of experimental research: manipulation of variables and control by random assignment (p. 1052). For this reason, the work of Ato et al. (2013) in our study.

11. The sampling in line 154-155 What are your rational/methods to determine your sampling (295 social sciences and law (53.6%), and 224 engineering and architecture (46.4%)? Method should be clear and logical.

• Authors: In the validation process of the tool, the participation of 2 different areas of knowledge was considered, making it possible to verify the factorial invariance. Thus, in selecting students from these 2 specific subject areas, we have tried to ensure that they are representative of both humanities/social studies and science/technology, and also that the sample sizes allow for the indicated analysis.

12. line 148 Materials and method- what does materials mean for you? Have you used any material in conducting this study? My answer is no? Then better to say Methods only (if you use laboratory/experimental research you need to use materials where you have used human beings as research participants, that is all).

• Authors: We thank the reviewer for the suggestion. The "Materials and Methods" section follows the indications of PLOS ONE collected in the "Organization of the Manuscript", within the "Submission Guidelines". From our point of view, the information on materials is oriented to the instruments used, which in the case of this research, is an evaluation tool, specifically a questionnaire. However, following the reviewer's suggestion, we have changed the name of the section.

13. Even line 159 "Measures" do you think appropriate here? I think instruments seem more sound because they are accustomed to most social and behavioural research.

• Authors: We thank the reviewer for reflecting on this aspect. Continuing with the justification of comment number 12, the "Submission Guidelines" of PLOS ONE indicate the following about the "Material and Methods" section:

The Materials and Methods section should provide sufficient detail to allow properly trained investigators to fully replicate your study. Specific information and/or protocols for new methods should be included in detail. If the materials, methods, and protocols are well established, authors may cite articles where those protocols are described in detail, but the presentation must include enough information to be understood independently of these references.

However, in line with reviewer comment #12, we followed their suggestion and changed the name of the section

---

## [Editor Report · Decision Letter 1]

27 Jul 2022

EXECUTIVE FUNCTION ASSESSMENT: ADAPTATION OF THE AMSTERDAM EXECUTIVE FUNCTION INVENTORY USING SPANISH FIRST-YEAR UNIVERSITY STUDENTS FROM TWO KNOWLEDGE AREAS

PONE-D-22-11155R1

Dear Dr. Romero-Galisteo,

We’re pleased to inform you that your manuscript has been judged scientifically suitable for publication and will be formally accepted for publication once it meets all outstanding technical requirements.

Kind regards,

Gabriel G. De La Torre

Academic Editor

PLOS ONE

---

## [Editor Report · Acceptance letter]

10 Aug 2022

PONE-D-22-11155R1 

EXECUTIVE FUNCTION ASSESSMENT: ADAPTATION OF THE AMSTERDAM EXECUTIVE FUNCTION INVENTORY USING SPANISH FIRST-YEAR UNIVERSITY STUDENTS FROM TWO KNOWLEDGE AREAS 

Dear Dr. Romero-Galisteo:

I'm pleased to inform you that your manuscript has been deemed suitable for publication in PLOS ONE. Congratulations! Your manuscript is now with our production department. 

Kind regards, 

on behalf of

Dr. Gabriel G. De La Torre 

Academic Editor

PLOS ONE